# ORFeome Phage Display Reveals a Major Immunogenic Epitope on the S2 Subdomain of SARS-CoV-2 Spike Protein

**DOI:** 10.3390/v14061326

**Published:** 2022-06-17

**Authors:** Rico Ballmann, Sven-Kevin Hotop, Federico Bertoglio, Stephan Steinke, Philip Alexander Heine, M. Zeeshan Chaudhry, Dieter Jahn, Boas Pucker, Fausto Baldanti, Antonio Piralla, Maren Schubert, Luka Čičin-Šain, Mark Brönstrup, Michael Hust, Stefan Dübel

**Affiliations:** 1Institut für Biochemie, Biotechnologie und Bioinformatik, Abteilung Biotechnologie, Technische Universität Braunschweig, Spielmannstr 7, 38106 Braunschweig, Germany; f.bertoglio@tu-braunschweig.de (F.B.); s.steinke@tu-bs.de (S.S.); p.heine@tu-bs.de (P.A.H.); maren.schubert@tu-bs.de (M.S.); 2Helmholtz Centre for Infection Research, Inhoffenstr. 7, 38124 Braunschweig, Germany; sven-kevin.hotop@helmholtz-hzi.de (S.-K.H.); zeeshan.chaudhry@helmholtz-hzi.de (M.Z.C.); luka.cicin-sain@helmholtz-hzi.de (L.Č.-Š.); mark.broenstrup@helmholtz-hzi.de (M.B.); 3Institut für Mikrobiologie, Technische Universität Braunschweig, Spielmannstr. 7, 38106 Braunschweig, Germany; d.jahn@tu-bs.de; 4Institute of Plant Biology, Technische Universität Braunschweig, Humboldtstr 1, 38106 Braunschweig, Germany; b.pucker@tu-braunschweig.de; 5Department of Clinical, Surgical, Diagnostic and Pediatric Sciences, University of Pavia, 27100 Pavia, Italy; f.baldanti@smatteo.pv.it; 6Molecular Virology Unit, Microbiology and Virology Department, IRCCS Fondazione Policlinico, 27100 Pavia, Italy; a.piralla@smatteo.pv.it

**Keywords:** phage display, epitope mapping, COVID-19, genomic library, NGS

## Abstract

The development of antibody therapies against SARS-CoV-2 remains a challenging task during the ongoing COVID-19 pandemic. All approved therapeutic antibodies are directed against the receptor binding domain (RBD) of the spike, and therefore lose neutralization efficacy against emerging SARS-CoV-2 variants, which frequently mutate in the RBD region. Previously, phage display has been used to identify epitopes of antibody responses against several diseases. Such epitopes have been applied to design vaccines or neutralize antibodies. Here, we constructed an ORFeome phage display library for the SARS-CoV-2 genome. Open reading frames (ORFs) representing the SARS-CoV-2 genome were displayed on the surface of phage particles in order to identify enriched immunogenic epitopes from COVID-19 patients. Library quality was assessed by both NGS and epitope mapping of a monoclonal antibody with a known binding site. The most prominent epitope captured represented parts of the fusion peptide (FP) of the spike. It is associated with the cell entry mechanism of SARS-CoV-2 into the host cell; the serine protease TMPRSS2 cleaves the spike within this sequence. Blocking this mechanism could be a potential target for non-RBD binding therapeutic anti-SARS-CoV-2 antibodies. As mutations within the FP amino acid sequence have been rather rare among SARS-CoV-2 variants so far, this may provide an advantage in the fight against future virus variants.

## 1. Introduction

The novel beta-coronavirus SARS-CoV-2 was described in late 2019 and is responsible for the current public health crisis of global concern [1,2,3,4]. To tackle this pandemic, it is important to understand which viral proteins are targeted by the humoral response mounted by the host organism. The wild type SARS-CoV-2 viral genome [5] encodes four structural, 16 non-structural and several accessory proteins within the 29,903 nucleotides (nt) of (+)ssRNA [6,7,8,9,10,11]. The pathogenicity of SARS-CoV-2 is linked to its ability to bind the human angiotensin-converting-enzyme 2 (ACE2) [12]. ACE2 is recognized by the receptor-binding-domain (RBD) of the S1 subunit of the spike protein. Upon binding, the spike protein mediates viral cell fusion by changing from its pre-fusion to its post-fusion condition [13,14]. Once the virus particle is attached to the host cell, the spike is processed by host cell proteases such as furine, TMPRSS2 or cathepsins. These enzymes TMPRSS2 prime the spike protein for efficient cell entry, leading to the infection of the host cell [15]. COVID-19 patients mount a significant immune response after infection or vaccination, including neutralizing antibodies [16,17]. Most therapeutic approaches using antibodies are focused on preventing the virus from binding to host cells. Targeting the RBD-ACE2 interface with such inhibiting antibodies has been a successful approach to generate approved therapeutics to prevent viral cell attachment [18,19,20,21,22]. However, newly arising SARS-CoV-2 variants have proven that the virus can introduce mutations within the RBD that lead to loss-of-function of known therapeutics [23,24,25]. Besides the spike protein, the nucleocapsid (N) protein is also highly immunogenic and is currently used in most point-of-care (PoC) coronavirus antigen tests available on the market [26]. Other viral proteins, such as ORF3a, that is involved in viral replication and release, or ORF8, which is an Ig-like folded dimer that is poorly conserved among coronaviruses and interacts with many host cell proteins, are currently under investigation to ascertain their suitability as targets for the development of therapeutics [7]. Despite the fact that cell binding relies on RBD, directly neutralizing anti-RBD antibodies may only be one part of anti-viral defenses, as other mechanisms such as CDC, ADCC or ADCP may help to clear the virus without direct interference with the RBD/ACE2 interaction [27]. Along these lines, antibodies targeting the N-terminal domain (NTD) of the spike protein showed neutralizing capacities [28]. In this respect, it is also interesting to analyze the role of antibodies on the epitopes outside of the RBD. In the past, the identification of immunogenic epitopes via phage display has been useful for the development of vaccines and passive immunization approaches. Using phage display, Riemer et al. [29] discovered that the short peptide sequences of the trastuzumab antigen Her-2/neu were immunogenic. This peptide has been declared a mimotope, because it mimics the parts of the epitope that are responsible for antibody binding, and has been successfully used to induce the production of highly specific antibodies, similar to trastuzumab in vivo in mice. They showed that these antibodies have a neutralizing effect. To identify such mimotopes for SARS-CoV-2, many research groups have performed assays based on microarray approaches. Here, synthetic peptides are used to determine the binding site of antibodies in patient sera. Potential immunogenic peptides have been identified for the spike protein using a set of synthetic, overlapping peptides. In a study by Wang et al. [30], two major immunogenic peptides were identified with this method, both representing fragments of the spike protein. One ranges from amino acid 561 to 579 just downstream the ACE2-RBD interface, while the other has been located on S2 between amino acids 818 and 835, i.e., the fusion peptide that is highly conserved among coronaviruses [30]. A comparable attempt by Li et al. [31] confirmed similar immunogenic epitopes on the spike protein. Such proteome analysis using peptide arrays is a common tool to identify immunogenic epitopes. However, the findings vary between studies depending on the method but also on the analyzed patient serum samples. In particular, peptide arrays are limited by the maximal length of the synthetic peptides (typically around 15–20 amino acids), thus missing conformational or many non-linear epitopes, which make up a significant fraction, i.e., around 40%, of antibody epitopes [32]. ORFeome display can detect immunogenic proteins independent of the transcriptome or proteome, as it does not rely on cDNA or protein extracts [33,34,35]. In this study, we present an ORFeome phage display approach with the aim of identifying immunogenic peptide sequences across the SARS-CoV-2 genome.

## 2. Materials and Methods

### 2.1. Patient Serum Samples

In total, 20 samples were obtained in Germany and Northern Italy between February 2020 and January 2021 from non-vaccinated patients that had been hospitalized at intensive care units (ICU). Sampling was performed according to the Declaration of Helsinki. Plasma samples were taken between March 2020 and February 2021. Approval for the serum samples was given from the ethical committee of the Technische Universität Braunschweig (Ethik-Kommission der Fakultät 2 der TU Braunschweig, approval number FV-2020-02) and from the Institutional Review Board of Policlinico San Matteo (protocol number P_20200029440).

### 2.2. Library Construction

The SARS-CoV-2 encoding sequences were separately amplified from seven genome fragments in pUC57 vectors via PCR using individual primers for each fragment. DNA of each fragment was purified using the Macherey-Nagel PCR purification Kit according to the manufacturer’s instructions. PCR products were sonicated for 30 s on/30 s off for 70 cycles (Bioruptor, Diagenode). The obtained fragmented DNA was blunt-end polished (Fast DNA Repair Kit, Thermo Fisher, Braunschweig, Germany), purified with the NucleoSpin Gel and PCR Clean-up Kit (Macherey-Nagel, Düren, Germany) and cloned into pHORF3 with T4 DNA Ligase (NEB) overnight at 16 °C followed by a heat-inactivation at 70 °C for 10 min [33]. Ligations were purified using Amicon Ultra 0.5 mL tubes (Merck Millipore, Tullagreen, Ireland) according to the manufacturer’s instructions. Electro competent *E. coli* SS320 (Lucigen, Middleton, WI, USA) were used for transformation in a pre-chilled 0.1 cm electroporation cuvette using a micropulser (BioRad, Munich, Germany) with 1.8 kV for at least 5 ms. Immediately after pulsing, 1 mL of pre-warmed recovery medium (Lucigen) was added and cells were recovered for one hour at 37 °C, 650 rpm. Next, 10 µL of transformed *E. coli* suspension were used for a dilution series ranging from 10^−4^ to 10^−6^ in 2xYT medium and plated them on 2xYT agar plates supplemented with 100 µg/mL Ampicillin and 0.1 M glucose (2xYT-GA). The next day, the obtained colonies were counted and the maximum library diversity was determined. The remaining 990 µL recovered *E. coli* were plated onto a 15 × 15 cm 2xYT-GA plate. After overnight cultivation, the clones were collected by applying 25 mL 2xYT-GA medium onto the plates and incubation for 15–30 min on a plate rocker. The suspension was harvested with a L-spatula and 1 mL aliquots were saved as glycerol stocks (750 µL medium and 250 µL 80% glycerol solution) by freezing them in liquid nitrogen and storing them at −80 °C.

### 2.3. Colony PCR for Library Quality Control

*E. coli* cells that contained a pORF3 vector were analyzed with PCR according to the method described by Kügler et al. [33]. Briefly, the PCR was carried out with the primers MHLacZPro_f (5′-GGCTCGTATGTTGTGTGG-3′) and MHgIII_r (5′-CTAAAGTTTTGTCGTCTTTCC-3′) according to the following protocol: 1 min denaturation at 95 °C, 30 s at 56 °C for annealing and 1 min at 72 °C for elongation. The process was repeated 29 times with a final elongation step for 5 min. The PCR products were analyzed on a 1% agarose gel which ran at 120 V for 30 min in 1× TAE buffer.

### 2.4. ORF Enrichment with Hyperphage

To enrich SARS-CoV-2 related protein fragments, the library was packaged using Hyperphage (Progen, Heidelberg, Germany) that lacks the gene encoding pIII [36]. To do so, the obtained glycerol stocks were inoculated in 200 mL 2xYT-GA medium. The bacterial culture grew at 37 °C and 210 rpm until it reached an optical density at 600 nm (OD600) of approximately 0.5. Then, 25 mL of the culture was infected with 2.5 × 10^11^ cfu of Hyperphage. The culture incubated at 37 °C without shaking for 30 min and at 37 °C with shaking for 30 min at 250 rpm. Cells were harvested by centrifugation at 3220× *g* for 10 min and resuspended in 2xYT-KA medium for overnight phage production at 30 °C, 210 rpm. The next day, the culture was centrifuged for 30 min at 8000 rpm (Sorvall Centrifuge RC 6+; Rotor F9S 4×000Y). The supernatant was harvested and 1/5 (*v*/*v*) of 20% PEG (*v*/*v*) in 2.5 M NaCl were added. The suspension was incubated overnight on ice in a 4 °C environment to precipitate phage. Next, the suspension was centrifuged for 1 h at 11,000 rpm. The supernatant was discarded and the remaining pellet was resuspended in 10 mL phage dilution buffer (PDB). The phage suspension was added to 1/5 of the final volume of 20% PEG (*v*/*v*) in 2.5 M NaCl and incubated overnight on ice. Then the suspension was centrifuged for 30 min at 20,000 rpm and the supernatant was discarded. The phage pellet was diluted in 1 mL PDB and added to a 2 mL Eppendorf tube. Finally, the cells were centrifuged 2× at 18,000× *g* and filtered in a 2 mL vial using a 0.45 µm filter. The libraries were stored at 4 °C. Libraries were titrated in a dilution series ranging from 10^−8^ to 10^−12^ and 10 µL of each dilution was used to infect 50 µL *E. coli* XL-1 Blue MRF’ cells (OD600 = 0.5). The infected bacteria were plated out on 2xYT-GA plates. 

### 2.5. NGS Data Analysis

For Next-Generation-Sequencing (NGS), the libraries were used to re-infect 5 mL of *E. coli* XL1-Blue (MRF’) cells at an OD600 = 0.4–0.6. The infected bacterial culture was inoculated overnight at 37 °C, 250 rpm and used for plasmid preparation with the Macherey-Nagel Easy Pure kit according to the manufacturer’s instructions. The obtained plasmids were used for PCR amplification of the sequence of interest. Here, it was important to introduce barcode sequences via primers for NGS analysis at GENEWIZ, according to the GENEWIZ Amplicon-seq guidelines. The PCR products were purified and analyzed on a 1% agarose gel. Resulting FASTQ files were mapped to the Wuhan reference genome (Genbank No.: MT326090.1) using BWA-MEM with the UGENE workflow “Processing of raw DNA-seq paired-end reads”. The resulting BAM file was further processed using samtools [37] and a coverage file was created according to a previously published python script which indicated the reads that cover each position on the SARS-CoV-2 genome [38]. The coverage file was plotted with OriginPro 2018. Furthermore, the average coverage of each ORF on the nucleotide level was analyzed. The calculated average value of the reads per position were plotted using OriginPro. 

### 2.6. Identification of Immunogenic Epitopes

The panning procedure was adapted from Zantow et al. [34]. Briefly, in the first pre-incubation step, the phage binding antibodies present in the sera were excluded from the library. To achieve this, three wells per serum of a 96-well microtiter plate (MTP) (High Bind, Corning, Glendale, AZ, USA) were coated with Hyperphage (10^11^ phage/well) and incubated overnight at 4 °C. In parallel, a 100-fold excess (diversity/library titer) of each library was pre-incubated for one hour at RT in panningblock solution (1% (*w*/*v*) BSA and 1% (*w*/*v*) milk powder in PBS + 0.05% (*v*/*v*) Tween20 on an unrelated human IgG antibody and on the anti-human Fc specific antibody MC002-M (Abcalis, Braunschweig, Germany), used for capturing of the sera. Serum antibodies dissolved in panningblock were captured by the anti-human Fc specific antibody MC002-M for two hours at RT. After washing, the pre-incubated libraries were applied and incubated two hours at RT. Unbound phage were washed away and 150 µL/well trypsin (10 µg/mL in PBS) was applied to each well for 30 min at 37 °C for elution. Then, 100 µL of eluted phage were used for infection of 5 mL *E. coli* TG1 cells at an OD600~0.5 in a 24-well MTP. The plates were incubated 30 min without and then 30 min with shaking at 450 rpm at 37 °C in a shaker (Labnet Vortemp56, Marshall Scientific, Hampton, NH, USA). Medium was changed by centrifuging the plate 10 min at 2500× *g* (Eppendorf 5810R) and drying the plate briefly on a paper towel. Afterwards, 5 mL of 2xYT medium supplemented with 100 µg/mL Ampicillin and 0.1 M glucose (2xYT-GA) were added to the pellet and incubated 30 min at 37 °C, 450 rpm. The suspension was infected with 17 µL Hyperphage (3 × 10^12^ cfu/mL) and incubated 30 min at 37 °C without and afterwards 30 min at 37 °C with shaking at 450 rpm. Plates were centrifuged as before and 5 mL 2xYT-A + 70 µg/mL Kanamycin (2xYT-KA) were added to the pellet. Phage production was carried out overnight at 30 °C, 450 rpm. For the next round of panning, 100 µL produced phage were added to 50 µL panningblock solution and applied to the pre-incubation steps. After the second panning round, 1:10, 1:100 and 1:1000 dilutions of the eluted phage were prepared in 2xYT medium. From each dilution, 10 µL were used to infect 50 µL *E. coli* XL1-Blue MRF’ (OD600~0.5). The infected bacteria were plated on 2xYT-GA agar plates. Randomly selected clones were analyzed as indicated above and alignments were done with UGENE (version 34). 

### 2.7. Phage Clone Production for Screening

For each panning approach, a 96-well MTP was filled with 150 µL 2xYT-GA medium per well after the third panning round. Individual clones were picked from the 2xYT-GA agar plates and inoculated in the 96-well MTP overnight at 37 °C, 800 rpm. This plate served as a master-plate and was stored at −80 °C in 20% (*v*/*v*) glycerol. The next day, 10 µL were transferred from each well to 180 µL 2xYT-GA in a new 96-well MTP. The MTP was incubated two hours at 37 °C, 800 rpm before it was infected with 10 µL/well of Hyperphage (3 × 10^12^ cfu/mL). After incubation for 30 min at 37 °C with and without shaking at 800 rpm, the plate was centrifuged at 3220× *g* for 10 min. The pellets were resuspended in 2xYT-KA medium and phage particles were produced overnight at 30 °C, 800 rpm. For epitope mapping of monoclonal scFv-Fc antibodies, a single panning round was performed.

### 2.8. Screening ELISA

For screening ELISA experiments, 96-well MTP (High Bind, Corning, Glendale, AZ, USA) were coated with 25 ng/well of a mouse anti-pVIII antibody (clone B62-FE2; Progen) and incubated overnight at 4 °C. Wells were washed three times with Milli-Q + Tween20 (0.1% (*v*/*v*)) and blocked with 350 µL/well of 2% (*w*/*v*) M-PBST. Overnight produced phage particles were captured and COVID-19 patient sera were diluted 1:100 in 2% M-PBST supplemented with 10^10^ cfu Hyperphage and 1:10 (*v*/*v*) *E. coli* cell lysate and applied to the captured phage particles. To detect bound serum antibodies, an anti-human IgG polyclonal antibody conjugated with HRP (A0170; Sigma Aldrich, St. Louis, MO, USA) was applied at a dilution of 1: 70,000 and incubated two hours at RT. The plates were washed three times with Milli-Q + Tween20 and TMB substrate was applied for 20 min to each well. After stopping the reaction with 1 N sulfuric acid the absorption at 450 nm and 620 nm was measured. In the assays using monoclonal antibodies, clones were considered positive with a ΔOD450-620 nm signal higher than 1.3.

### 2.9. Epitope Characterization by Peptide Microarray

Microarray assays were carried out according to the method described by SK-Hotop et al. [39]. Briefly, 15-mer peptides of the four structural SARS-CoV-2 proteins (Ref. NC_045512, Wuhan-Hu-1) with a three amino acid overlap were synthesized on cellulose membranes, dissolved and printed to glass slides. Patient serum samples were diluted 1:120 in blocking buffer (2% casein in TTBS (1% Tween 20 (*w*/*v*) in TBS) and incubated on the slides over night at 4 °C. Bound serum antibodies were stained with Alexa Fluor^®^ 647 conjugated, isotype specific secondary antibodies (Jackson ImmunoResearch, West Grove, PA, USA). Data analysis was performed by visual inspection.

### 2.10. Titration ELISA on Identified Peptides

Biotinylated peptides were obtained from Peps4LS (Heidelberg, Germany). Costar 384-well MTP were coated with 200 ng/well Streptavidin overnight at 4 °C. The following day, all wells were washed three times with Milli-Q + Tween20 and blocked for one hour at RT with 2% (*w*/*v*) BSA dissolved in PBS-T. Next, 1 µg of biotinylated peptide per well was captured for one hour at RT and plates were washed three times as described before. Sera were titrated in 2% BSA in PBS-T ranging from the dilution factor of 1:300 to 1:3 × 10^−8^ and applied to a 384-well MTP. Unbound serum antibodies were washed away three times with Milli-Q + Tween20. To detect bound serum antibodies, we applied an anti-human IgG polyclonal antibody conjugated with HRP (A0170; Sigma; 1:70,000) and incubated for one hour at RT. The plates were washed three times with Milli-Q + Tween20 and TMB substrate was applied for 20 min to each well and analyzed as described before. Data analysis and graphical plots were carried out using OriginPro (version 2018).

## 3. Results

### 3.1. The Genome of SARS-CoV-2 Is Represented within the ORFeome Phage Display Library

We constructed a genomic ORFeome display library covering the entire viral genome. By using the phagemid vector pHORF3, which selects for open reading frames (ORF) upon packaging with Hyperphage, phages display viral protein fragments of different sizes on their surface [36,40]. The alignment of the FASTQ files obtained from NGS to the Wuhan reference genome (Genbank No.: MT326090.1) showed a coverage of the entire SARS-CoV-2 genome (Figure 1A). The plotted coverage file indicated that the reads were distributed over the genome, but each nucleotide position was represented differently: the structural ORFs, such as Spike, Envelope (E), Membrane (M) and Nucleocapsid (N), were covered better than the accessory proteins. Overall, the library covered the spike gene with a median of 3849 reads per nucleotide position (Figure 1B). The 3′-end of the genome, where all the smaller non-structural proteins are encoded, showed less abundant coverage compared to ORF1ab. The least represented ORF was ORF6, since the median read count per nucleotide position was 8. In contrast, the N protein showed median coverage of 17,293 reads per nucleotide position. Taken together, the median of the coverage rate for each nucleotide position of the entire genome was 4310 reads.

### 3.2. Epitope Identification of Monoclonal Antibodies

To determine whether the generated SARS-CoV-2 genomic ORFeome library could be used to identify antibody epitope, we tested the non-neutralizing monoclonal antibody STE73-6C10 that was shown to recognize a peptide sequence within the S1 subunit [41]. The ORFeome phage display and subsequent screening ELISA results led to the identification 24 clones that were sequenced. Twenty-three of these clones were different, with insert lengths between 31 and 77 aa. All clones shared a minimal epitope region of 25 aa (CTEVPVAIHADQLTPTWRVYSTGSN) that is identical to a sequence on the S1 subunit of the spike protein (Genbank No.: MT326090.1) (Figure 2A). These findings were verified by peptide microarray analysis of STE73-6C10. Here, it was shown that the antibody recognized three peptides on the used slides (Figure 2B). Notably, peptides 207 and 208 showed a high signal, whereas peptide 209 was faintly recognized. Due to this decrease in binding of peptide 209, we determined a 12 amino acid long core epitope consisting of the motif VAIHADQLTPTW for antibody STE73-6C10.

### 3.3. Immunogenic Epitope in the S2 Subdomain of the Spike Protein Is Common among COVID-19 Patients

Five COVID-19 patient sera from Northern Italy and Germany were panned against the generated ORFeome library to determine the mimotopes of the antibodies generated during humoral response. Prior to ORFeome display, antibody titers against spike and RBD were determined (Appendix A). Patient TUBS21 revealed the highest IgG titer among the selected patient sera. A major immunogenic epitope within the S2 subdomain of the spike protein was clearly evident after two panning rounds, as indicated in Table 1. Screening ELISA results determined the amount of clones sent to sequencing. The sequenced clones were aligned to the spike protein’s aa sequence (Genbank No.: MT326090.1) and plotted with Ugene (Version 34) or Geneious (v8.1) (Appendix A). Alignment analysis of patient 17 revealed that this patient has developed antibodies against the N-terminal part of the N protein (Table 1). For patient 1, we identified two distinct mimotopes at the S2 subdomain of the spike protein. The main mimotope refers to the fusion peptide (FP), which bears the cleavage site of the TMPRSS2 and is essential for the viral-host cell fusion process [12]. A less abundant mimotope was located between HR1 and HR2 than from a helical structure during cell fusion [42]. The analysis of patient 18 led to the identification of similar mimotopes. Here, two clones representing the site between HR1 and HR2 were identified, and 18 clones referred to the TMPRSS2 cleavage site epitope, which was also abundant in patient 1. Moreover, we identified two spike related epitopes for sample TUBS21. Here, one immunogenic epitope within the SD1 subdomain of spike, close to the RBD-ACE2 interface, was identified. Eleven distinct clones with a size ranging from 25 to 61 aa determined a MER of 17 aa (ESNKKFLPFQQFGRDIA). The second epitope was determined by 33 clones ranging from 16 to 70 aa in size. The 14 aa long MER was defined between the aa811 to aa824 (KPSKRSFIEDLLFN), which was also observed for patient sample TUBS42, further strengthening the results obtained with sera collected in Italy. Overall, the mimotope represented by this peptide (DPSKPSKRSFIEDLLFNKVTLADA), encompassing the FP, was the most common and abundant. Therefore, it was synthesized, and more patient samples were analyzed by ELISA, to determine whether there was binding directed to this mimotope. As Figure 3A indicates, 12 out of 15 patients from Italy had generated antibodies recognizing this mimotope. Moreover, our peptide microarray results confirm that patient sera recognize, among others, this mimotope (Figure 3B).

## 4. Discussion

In this study, we identified an immunogenic polypeptide located within the spike protein of SARS-CoV-2 which is prominently recognized by COVID-19 patient sera. This peptide was found in four out of five serum samples from COVID-19 ICU patients by ORFeome phage display. The minimal epitope region (MER) encodes the 16 amino acid (aa) sequence (811-PSKRSFIEDLLFNKVT-828) and refers to the N-terminus of the fusion peptide (FP) of spike, which is involved in viral cell entry after the spike protein is cleaved by the host protease TMPRSS2 [12,43]. This result was further strengthened by the presence of antibodies directed to this peptide sequence in several other patient sera (15 out of 17) and confirmed by peptide microarray assays. Here, the analyzed patient sera could detect 15-mer peptides with an offset of three amino acids. We detected the presence of antibodies, recognizing the aa sequence of the FP and peptide sequences referring to the S1 subunit (553-ESNKKFLPFQQFGRDIA-571) of spike, which have also been found by others with different methodologies. [30,44,45]. Consistent with results described by Zamecnik et al. [46] we found similar immunogenic peptides on the spike protein and Nucleocapsid (N). Using ReScan with a phage display library spanning the SARS-CoV-2 proteome with 38 aa long fragments and a 12 aa overlap, they identified two epitopes that also represent the FP (residues 799-836; 818-855; RefSeq.: NC_045512). Furthermore, they also characterized an immunogenic epitope between residues 1141 and 1178 which we observed for patient 1 and patient 18 as well [46]. A similar approach using phage display led to the successful identification of immunogenic peptides representing the Zika virus envelope protein of infected patients. However, the limitation of phage display-derived methods is that binding to epitopes that rely on post-translational modifications or quaternary structures is unlikely to be detected [47].

In our study, we show that the identification of this epitope is possible using ORFeome phage display of a whole-genome SARS-CoV-2 library. The library quality determines whether immunogenic epitopes can be found within serum samples or not. To this end, our library was analyzed by NGS; the results indicated that the majority of the genome sequence was covered by the library, despite some accessory proteins being only marginally represented. For additional control, we showed that monoclonal antibody STE73-6C10 [41] binds its epitope on the SD2 subdomain of the spike protein (CTEVPVAIHDQLTPTWRVYSTGSN). The identified mimotope was confirmed by a peptide microarray assay, where we identified the 12 aa-long motif VAIHADQLTPTW. Compared to what we found with ORFeome phage display (31–72 aa), this peptide is much shorter, which shows that functionally displayed polypeptides on pIII of our SARS-CoV-2 genome library are, in general, longer than the used 15-mers in the peptide microarray assay.

Newly emerging SARS-CoV-2 variants can escape the immune response to earlier variants by introducing mutations in the RBD [23,24]. Targeting the FP with antibodies may prevent the host protease TMPRSS2 from cleaving the S2 subdomain, thus reducing viral cell entry [12]. Moreover, the aa sequence of FP does not vary between currently known SARS-CoV-2 spike variants [48]. The generation of antibodies against the FP could therefore lead to the identification of a candidate that can still be used against newly emerging SARS-CoV-2 variants. Small molecules targeting TMPRSS2 have been shown to prevent viral cell entry in vitro, even though the physiological function of TMPRSS2 remains elusive [49]. It remains unclear what effects could occur when targeting TMPRSS2 alone. It seems that this protease is also involved in influenza infection by cleavage of hemagglutinin [50]. Targeting the FP with specific antibodies may also offer an alternative approach, avoiding possible deleterious effects caused by the targeting of a host endogenous protein. The FP leads to an immune response in COVID-19 patients and, to our knowledge, is not affected by N-gylcosylations [51]. This immunogenic character may be used to artificially initiate an immune response with this peptide sequence as part of a vaccine. Riemer et al. showed that short peptide fragments, i.e., 10 aa in size, can successfully induce an immune response for other diseases in a vaccine-like manner [29]. The FP sequence could have similar potential and might serve as a vaccine candidate that can induce the generation of SARS-CoV-2 eliminating antibodies. Our findings suggest that the spike protein’s FP of SARS-CoV-2 is an immunogenic sequence that is prominently targeted by the immune system of COVID-19 patients.

We showed that ORFeome phage display could identify this epitope in a fast and effective manner. The sequences may be useful for the development of therapeutics (either neutralizing antibodies or vaccines) which are more likely to be active against future variants, based on the observation that the FP sequence has remained conserved so far among SARS-CoV-2 circulating variants.

## Figures and Tables

**Figure 1 viruses-14-01326-f001:**
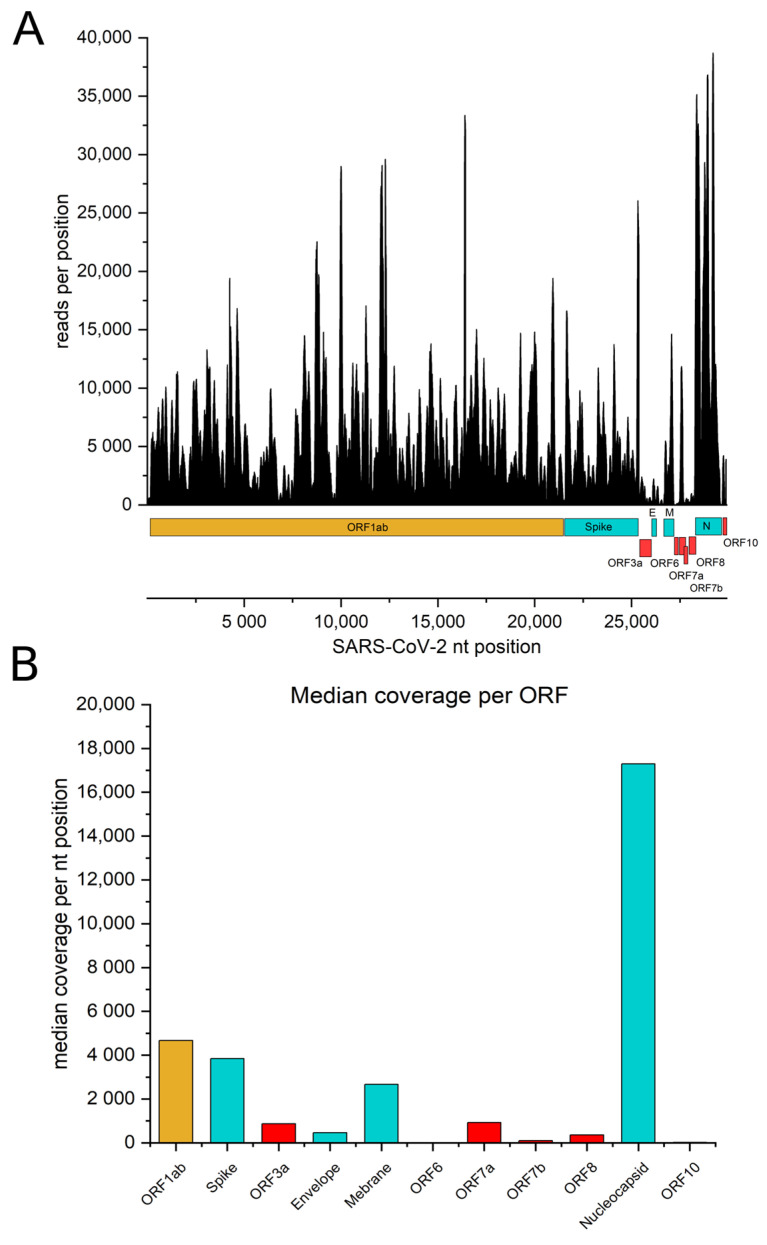
NGS data analysis. (**A**) NGS reads per nucleotide position plotted for each nucleotide position of the SARS-CoV-2 genome (Wuhan variant; Genbank No.: MT326090.1). A higher signal referred to a better coverage of the corresponding nucleotide position. (**B**) average reads per nucleotide position (as indicated in (**A**)) plotted against the length of the corresponding ORF, indicating the quality of coverage for each ORF. Structural ORFs are indicated in blue and ORFs encoding accessory proteins are indicated in red.

**Figure 2 viruses-14-01326-f002:**
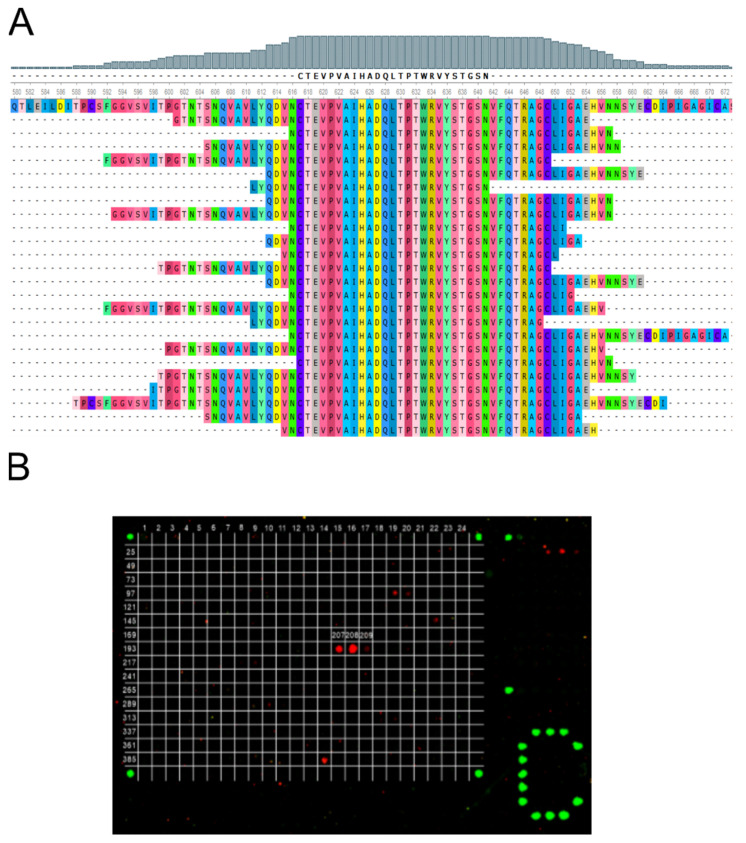
(**A**) alignment results of ORFeome phage display clones selected on STE73-6C10 scFv-hFc. Top sequence: spike of Wuhan variant, Genbank No.: MT326090.1. (**B**) Peptide microarray results of STE73-6C10 on the spike protein. The antibody recognizes the peptides 207-209 which correspond to the amino acid sequence 619–639 of the Wuhan variant, Genbank No.: MT326090.1.

**Figure 3 viruses-14-01326-f003:**
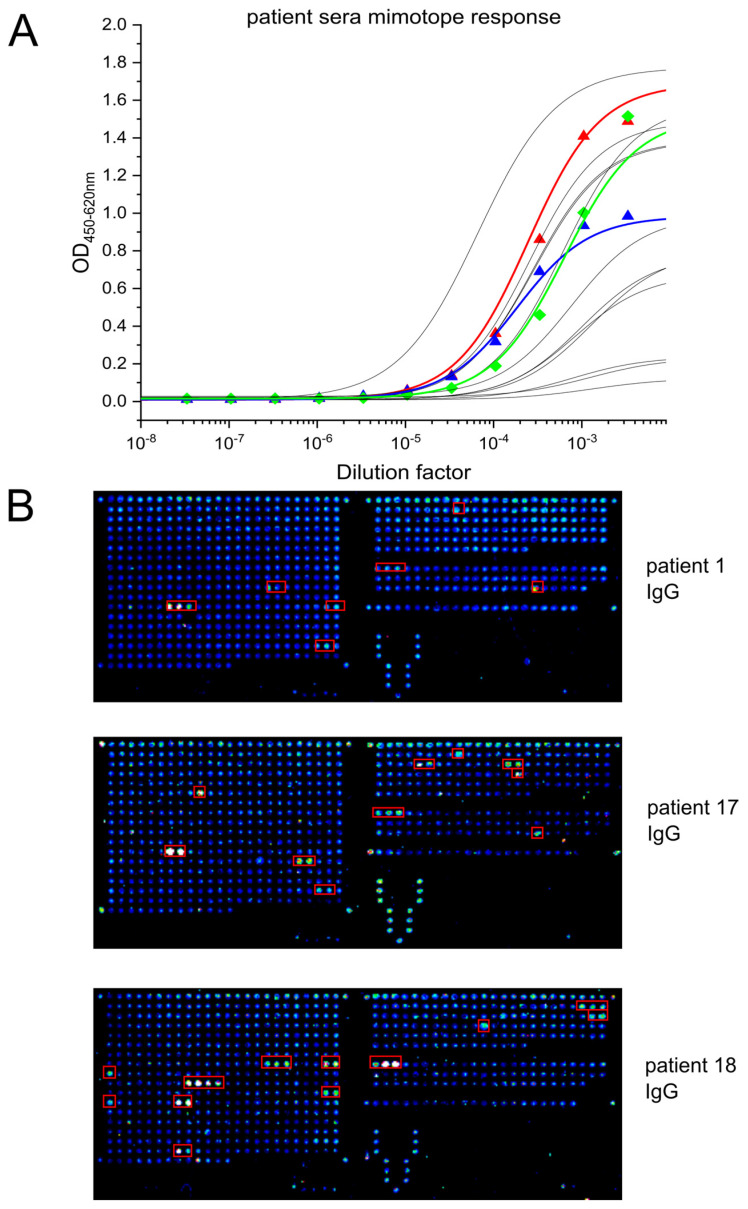
(**A**) Titration of serum samples from Italian patients on the identified mimotope synthetic peptide DPSKPSKRSFIEDLLFNKVTLADA. The serum samples used in ORFeome phage display panning are indicated with colors (red: patient 1, blue: patient 17 and green: patient 18). Patient samples from northern Italy that were not subject of ORFeome phage display are indicated in grey. Fitted curves were obtained by the Logistic5 function in OriginPro2018. (**B**) Peptide microarray analysis of the IgG response of patient sera 1, 17 and 18 on the four structural SARS-CoV-2 proteins. Left: Spike; Upper right: Nucleocapsid; Middle right: Membrane; Lower right: Envelope. Areas marked in red correspond to identified hits for linear epitopes detectable. Hit identification was carried out by visual inspection.

**Table 1 viruses-14-01326-t001:** Mimotopes of SARS-CoV-2 positive patients identified by ORFeome phage display. Amino acid positions are given according to Wuhan reference genome (Genbank No.: MT326090.1). Hits were determined by screening ELISA and clones that led to a higher signal than background were sent to sequencing.

Sample	Selected Hits	Epitope Sequence
patient 1	5/8	811-PSKRSFIEDLLFNKVT-828
	1/8	1143-ELDSFKEELDKYFKNHTSPDV-1165
patient 17	1	49- ASWFTALTQHGKEDLKFPRGQGVPINTNSSPDDQIGYYRRATRRIRGGDGKMKDLSPRWYFYYLGTGPEAGLPYGAN KDGIIWVATEGALNTPK DHIGTRNPANNAAIVL QLPQGTTLPKGFYAEGS -177
patient 18	18/22	811-PSKRSFIEDLLFNKVT-828
	2/22	1143-ELDSFKEELDKYFKN HTSPDVDLGDISGINASVVNIQKEIDR -1186
TUBS 21	11/67	553-ESNKKFLPFQQFGRDIA-571
	33/67	810-KPSKRSFIEDLLFN-825
TUBS 42	18/29	809-SKPSKRSFIEDLLF-824

## Data Availability

Not applicable.

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
