# Peer review of "ORFeome Phage Display Reveals a Major Immunogenic Epitope on the S2 Subdomain of SARS-CoV-2 Spike Protein"

_viruses, 2022, doi:10.3390/v14061326_

Round 1

Reviewer 1 Report

The paper focuses on the use of ORFeome method to identify COVID-19 related mimotopes. The study was well designed and presented. The authors were able to identify key epitopes surrounding the fusion protein of the spike as being an immunodominat epitope. 

Suggestions:

1. Line 97: Would be good to list how many samples were collected.

2. Line 179: 1011 , please use superscript for 10 power 11.

3. Line 182: % of Tween.?

4. Line 244: How much peptide was used?

5. Line 298: 5 sera samples were used for analysis. What was the criteria to select this 5 samples. In line 320 indicated that 15 samples were available. Any particular reason why only panning with 5 samples were done? Also, does the authors know the antibody tigers of the 5 samples which would indicate the antibody response of the patients that were used.

Reviewer 2 Report

COVID19 is an important health issue. Although worldwide vaccination has had a remarkable effect on disease transmission and death, and limited significantly the spread of the virus, the pandemic is not over. The arise of new variants may again lead to new health emergencies worldwide.

Here, the authors have used a phage display technique they developed (ORFome) to produce phage particles display fragments of the SARS-CoV2 proteome on the surface. They achieved this by fragmenting the viral genome and inserting into a bacteriophage genome, in frame with gene III (which codes for a minor surface protein). By panning on IgG from COVID19 patients, they have selected phage display epitopes recognized by antibody, which lead to identified epitopes recognized by 5 COVID patients. One this epitopes was further validated in larger cohorts and 15 out 17 patients reacted with it.

The study is interesting and likely to contribute for a better knowledge regarding the antibody response of these patients. The authors have mapped a major immunogenic epitope on the S2 subdomain, which may be involved in the processing of the spike protein by cell protease and, therefore, modulate viral cell entry. Further studies, however, are necessary to assess the importance of these finding in term of vaccine potential.

Minor issues:

a) the authors should describe with further details how the selection was performed and phage clones identified (Table 1). Was there enrichment of the selected phage with regard to other phage clones? For instance, for patient 1 it says 5 out of 8 hits while in patient 21 it says 11 of 67 hits. What does it mean (hit)? Number of phage clones sequenced? Why only 1 for patient 17, or 8 for patient 1 and 67 for the other patient (no. 21)? Table 1 is a bit confusing. 

b) please, include the alignments used to identify the minimal consensus epitope in table 1.

b) it would be helpful to map the epitopes on the structure of the spike protein. Are they surface available? Are they affected by spike protein N-glycosylation?
